# Timeline of Adverse Events during Immune Checkpoint Inhibitors for Advanced Melanoma and Their Impacts on Survival

**DOI:** 10.3390/cancers14051237

**Published:** 2022-02-27

**Authors:** Lorena Villa-Crespo, Sebastian Podlipnik, Natalia Anglada, Clara Izquierdo, Priscila Giavedoni, Pablo Iglesias, Mireia Dominguez, Francisco Aya, Ana Arance, Josep Malvehy, Susana Puig, Cristina Carrera

**Affiliations:** 1Melanoma Group, Institut d’Investigacions Biomediques August Pi I Sunyer (IDIBAPS), 08036 Barcelona, Spain; lorevc_63@hotmail.com (L.V.-C.); podlinik@clinic.cat (S.P.); pgiavedo@clinic.cat (P.G.); piglesia@clinic.cat (P.I.); midoming@clinic.cat (M.D.); jmalvehy@clinic.cat (J.M.); spuig@clinic.cat (S.P.); 2Medicine Department, Medicine Faculty, Campus Clínic, University of Barcelona, 08036 Barcelona, Spain; nanglada@gmail.com (N.A.); cizquierdo95@gmail.com (C.I.); 3Medical Oncology Department, Hospital Clinic of Barcelona, University of Barcelona, 08036 Barcelona, Spain; faya@clinic.cat (F.A.); amarance@clinic.cat (A.A.); 4Biomedical Research Networking Center on Rare Diseases (CIBERER), ISCIII, 08036 Barcelona, Spain

**Keywords:** immunotherapy, melanoma, dermatological adverse events, immune-related adverse events, immune checkpoint inhibitors, dermatological drug reactions, survival, outcome

## Abstract

**Simple Summary:**

A cohort of 153 melanoma patients treated with immune checkpoint inhibitors as first line therapy were studied, to specifically describe the timelines of all adverse events by the target organ, demonstrating a different profile of appearance over time. Interestingly, the survival benefit of presenting immune-related adverse events was demonstrated only for dermatological events, but a multivariate analysis found that this benefit is no longer significant after adjusting for the duration of therapy and the baseline stage of disease. In our opinion, the apparent good response marker of adverse events needs to be analyzed, taking into account the time in therapy and other prognostic markers, such as disease burden.

**Abstract:**

Immune-related adverse events (irAEs) are frequent and could be associated with improved response to immune checkpoint inhibitors (ICIs). A prospective cohort of advanced melanoma patients receiving ICI as first-line therapy was retrospectively reviewed (January 2011–February 2019). A total of 116 of 153 patients presented with at least one irAE (75.8%). The most frequent irAEs were dermatological (derm irAEs, 50%), asthenia (38%), and gastrointestinal (29%). Most irAEs appeared within the first 90 days, while 11.2% appeared after discontinuation of the therapy. Mild grade 1–2 derm irAEs tended to appear within the first 2 months of therapy with a median time of 65.5 days (IQR 26-139.25), while grade 3–4 derm irAEs appeared later (median 114 days; IQR 69-218) and could be detected at any time during therapy. Only derm irAE occurrence was related to improved survival (HR 6.46). Patients presenting derm irAEs showed better 5-year overall survival compared to those with no derm irAEs (53.1% versus 24.9%; *p* < 0.001). However, the difference was not significant when adjusting for the duration of therapy. In conclusion: the timeline of immune-related-AEs differs according to the organ involved. The (apparent) improved survival of patients who present derm AEs during immunotherapy could be partially explained by longer times under treatment.

## 1. Introduction

Immune checkpoint inhibitors (ICIs) have transformed cancer treatment by providing clear survival benefits in a wide range of cancers, with an acceptable safety profile [1,2,3,4,5]. However, ICIs can cause excessive immune “invigoration” and a diverse spectrum of immune-related adverse events (irAEs) [6,7,8].

There have, in fact, been irAEs that have led to fatal events (mainly colitis, myocarditis, pneumonitis) or to the development of lifelong secondary immune disorders (diabetes, hypothyroidism) [9,10,11]. Recently, management algorithms have improved approaches to the most common irAEs, resulting in a reduction of serious toxicities and related deaths [9,12,13]. Currently, however, there are no biomarkers that are able to predict the occurrence of an irAE, and only a few studies have described intrinsic factors related to the patient or to the primary tumor that could lead to an increased risk of adverse events [7,14].

Knowing the pattern of appearance of irAEs is useful to improve the safety of the treatment and to set up early management of any complications [13,15,16,17].

Dermatological irAEs are considered some of the most frequent reactions during immunotherapy, but data regarding the temporality and influence on therapy management are scarce. Moreover, several studies found a possible role of irAEs as good response markers [18,19,20,21,22,23,24,25,26,27,28], where the presence of an irAE has been linked with multiple favorable cancer outcomes, including ORR, PFS, and OS [19,21,22]. The objective of this study was to characterize the timeline and clinical presentation of dermatologic irAEs and identify the impact of irAEs on survival.

## 2. Materials and Methods

A cohort study was conducted at the Hospital Clinic of Barcelona, Spain, from January 2011 to May 2019. Patients eligible for inclusion were those diagnosed with melanoma, classified as stages IIIC and IV, according to the 7th edition of the American Joint Committee on Cancer (AJCC) staging system and treated with immunotherapy as first line therapy. All included patients received at least one dose of immunotherapy and were followed up for at least three months. Patients were evaluated according to the results of clinical surveillance of advanced melanoma patients at least every three months with body scan CT imaging and laboratory tests, and additionally in cases where adverse events occurred.

Exclusion criteria were other lines of treatment—previous targeted treatment or chemotherapy—to prevent a delayed effect of the medication from being attributed to the immunotherapy. All patients gave their informed consent to be part of an online up-to-date safe database (Xarxa Catalano Balear de Centres de Melanoma). This study used an approved protocol in accordance with the provisions of regulation (EU) 2016/679 by the Board of Research Ethics of the hospital’s CEIM. This study was performed following the 2015 STROBE guidelines [29].

### 2.1. Adverse Events Registration Protocol

All irAEs were recorded and classified based on the affected organ. In addition, subtypes of adverse reactions were classified according to the organ/system and the degree of affectation according to the Common Terminology Criteria for Adverse Events of the National Cancer Institute Version CTCAE 5.01(2018). An investigative dermatologist was responsible for checking medical records and the consistency of the information provided, with respect to the characteristics of the irAEs. Since adverse events were evaluated retrospectively, in order to homogeneously collect the events, all were classified according to the CTCAE scale v5.01 (2018). The times at which skin reactions appeared were evaluated with respect to any other reactions, the drug associated with this reaction, and the patient’s condition at that time.

### 2.2. Statistical Analysis

Pearson’s X^2^ test and a trend test for ordinal variables were used to compare categorical and ordinal variables, respectively. For continuous variables, the Wilcoxon test was used for comparison between two groups of samples and the Kruskal–Wallis test for comparing multiple groups [30]. The baseline characteristics of the patients were summarized using the number and percentage for categorical variables, and the median and range for continuous variables, and we plotted the analysis using boxplot and density plots [31].

Survival curves based on Kaplan–Meier methods and a log-rank test were used to investigate differences in post treatment OS with respect to global and skin toxicities. Curves were calculated using the ‘survfit’ function in the ‘survival’ package and plotted with the ‘survminer” package in R [32,33,34].

Multivariate survival analyses were performed using Cox’s proportional hazards model. Models were fitted using the ‘coxph’ function in the ‘survival’ package in R [33,34]. Hazard ratio estimates were calculated for the effect of skin toxicities on OS adjusted for AJCC stages and the duration of treatment (as categorical variables stratified by quartiles).

All statistical tests were two-sided and *p* values ≤ 0.05 were considered statistically significant. All tests were performed using the computing environment R [35].

## 3. Results

Initially, 341 patients were identified with advanced melanoma, who had undergone systemic therapy. After applying the inclusion and exclusion criteria, a final cohort of 153 patients was included for analysis. The reasons for exclusion were: treatment other than immunotherapy initiated as the first line (*n* = 149), patients started treatment before 2010 (*n* = 9), and no follow-up performed (*n* = 30).

Patients had a median age of 60 years (interquartile range (IQR) 45–69) at the time of initiating immunotherapy. Globally, (66) 45.2% were women and, according to the AJCC 7^TH^ version, were stratified as 28 Stage IIIC (18.9%), 22 Stage IVA (14.9%), 28 Stage IVB (18.9%), and 70 Stage IVC (48.2%). Most tumors (75.8%) were *BRAF* wild type. The baseline characteristics of the patients, stratified by the presence and type of irAE, are shown in Table 1.

### 3.1. Adverse Events Profile

#### 3.1.1. Drug-Related AEs

Of the 153 patients included in the study, 116 (75.8%) developed an irAE during or within 3 months of finishing immunotherapy. Regarding the immune checkpoint drugs: an irAE occurred in 64.3% of patients receiving ipilimumab, 79% of pembrolizumab patients, 72.7% of the nivolumab group, and in 77.7% of those receiving the combination regimen ipilimumab plus nivolumab.

#### 3.1.2. Target Organ and Timeline of IrAEs

The analysis by target organ-adverse events shows that the most frequent involvement was the skin (50% of cases), followed by asthenia (38%), gastrointestinal (29%), and hepatobiliary (21%) (Figure 1A). Most irAEs appeared within the first 90 days of treatment and asthenia was the earliest event recorded (median time 42.5 days, IQR 21-104). Density and box plots (Figure 2A,B) show the differences according to the target organ: the onset time of asthenia, skin, gastrointestinal, and endocrine toxicities occurred mainly at the beginning of the treatment, while musculoskeletal and connective tissue toxicities could occur at any time and remained constant over time. The median time to the emergence of dermatological irAEs was 65.5 days (IQR 26-139), whereas for musculoskeletal and connective tissue irAEs, 187 days (IQR 70-316), *p* < 0.0001.

#### 3.1.3. Severity of IrAEs

A total of 9 patients (5.9%) presented 10 moderate/severe irAEs (grades 3–4 according to CTCA v.5). There were no significant differences between the occurrence of grade 3–4 irAEs and the culprit drug. The analysis of target organ irAEs showed that grade 3–4 toxicities manifested more frequently as hepatobiliary (10% of patients), gastrointestinal (7%), and cutaneous (6%) (Figure 1A). Moreover, density and box plots (Figure 2C,D) showed differences related to the median appearance time for grade 3–4 irAEs; median time to dermatological grade 3–4 irAEs was 114 days (IQR 69-218), while median time for gastrointestinal irAEs was 50 days (IQR 44-81), (*p* = 0.05). Importantly, dermatological and hepatobiliary irAEs could be detected at any time during treatment, in contrast to gastrointestinal irAEs, which tended to develop earlier than the other two.

### 3.2. Dermatological Adverse Events (Derm IrAEs)

Of the 153 patients included in the study, 76 (50%) developed dermatological irAEs during the treatment with immunotherapy. Derm irAEs presented a wide spectrum of disorders (Figure 1B). Overall, the most frequent derm irAEs seen were pruritus in 47 patients (31%), maculopapular rash in 42 (28%), eczematous dermatitis in 11 (7%), vitiligo-like depigmentation in 11 (7%), and lichenoid mucositis in 7 patients (5%).

Moderate or severe toxicities (grade 3–4) were observed in 9 patients, and 1 presented two severe derm irAEs simultaneously (Figure 1B shows all derm irAEs observed). A total of 11.8% of derm irAEs were considered grades 3 or 4.

The most severe derm irAEs consisted of generalized maculopapular rash (3%), erosive lichenoid mucositis (2%), DRESS syndrome (1%), and erythema multiforme (1%).

Mild grade 1–2 derm irAEs tended to appear within the first 2 months of therapy with a median time of 65.5 days (IQR 26-139.25), while grade 3–4 derm irAEs appeared later (median 114 days; IQR 69-218) and could be detected at any time during therapy. Importantly, most of the irAEs that appeared after discontinuing therapy were dermatological (85% of delayed onset irAEs).

### 3.3. Survival Analysis

Survival analysis with Kaplan–Meier plots showed no significant improvement in overall survival of those patients who contracted an irAE during treatment. However, the stratified analysis by the target organ of toxicity showed that patients who presented derm irAEs presented a statistically significant survival advantage compared to those who did not (*p* < 0.001). Patients who developed derm irAEs with immune checkpoint therapy presented a five-year overall survival of 53.1% (95% CI, 37.7–74.8%) versus 24.9% (95% CI, 13.5–46.2%) of those who did not present derm irAEs (Figure 3).

A univariate Cox regression analysis showed a hazard ratio of 0.40 (0.24–0.66, *p* < 0.001) in the group of patients who presented derm irAEs. A univariate analysis demonstrated that the baseline clinical–pathological status (AJCC staging) and the duration of immunotherapy were also related to improved survival (HR 6.46 (2.57–16.23, *p* < 0.001) for stage IVC and HR 0.53 (0.32–0.88, *p* = 0.014 for those treated longer than 65 days)).

A multivariate analysis of the benefit of experiencing derm irAEs in overall survival, after adjusting the model for the AJCC stages and for the duration of the therapy, demonstrated that this survival advantage was no longer statistically significant (HR 0.74, 0.44–1.25, *p* = 0.263) (Table 2).

## 4. Discussion

This study into the development of immune-related adverse events (irAEs) during melanoma treatment with ICI demonstrates a specific timeline of presentation, depending on the organ involved and the severity of symptoms. In line with the literature [36,37], derm irAEs are the most frequent and can be detected in about 50% of patients, during treatment or even 3 months after ICI therapy is discontinued. Most patients will present derm irAEs during the first 3 months, similar to gastrointestinal and endocrine irAEs. It is thought that the majority of irAEs, including grade 3–4 events, are developed within 12 weeks of the initial administration [38]. However, certain irAEs can present a distinctive timeline: the most severe derm irAEs (grades 3–4), despite their infrequency (6% of patients), could appear at any time during follow-up, as with hepatobiliary irAEs, but in contrast to gastrointestinal severe irAEs, which present almost exclusively during the first 3 months. Musculoskeletal irAEs are rare and both mild and severe cases may be detected at any time during the therapy.

The second important result is that patients presenting derm irAEs showed a significantly better outcome compared to those with no derm irAEs (5-year overall survival, 53% vs. 25%). However, this apparent benefit in survival was not found after adjusting for the clinical baseline status and for the duration of therapy. Previous studies suggested that patients developing derm irAEs may experience longer intervals without disease progression, and a meta-analysis found that vitiligo-like depigmentation could be a biomarker of good response [1,18,20,21,23,24,25,26,27,28,39,40,41]. However, in view of our results, this apparent benefit could also be caused by a longer time under treatment: the longer the treatment, the better response the patient presents, and the more likely the irAEs will occur. In our opinion, the association between derm irAEs and outcome should be studied, and other prognostic markers that could influence this association should always be adjusted for.

On the one hand, it is reasonable to expect a more intense immune “invigoration” in such responses and, consequently, a higher likelihood of irAEs. On the other hand, it is well known that the lower the disease burden at baseline, the better the response to immunotherapy [42]. Moreover, this can be a long-term response leading to improved survival, and indeed, the time to withdraw ICI drugs is still under debate [43]. In the same line, it is reasonable to expect that those patients with better baseline statuses and who respond to treatment will receive long-term immunotherapy. The question is whether this population is the most likely to present irAEs as well. The specific timeline of derm irAEs could explain why this is not the case for gastrointestinal or endocrine toxicities, as they tend to appear within the first 3 months.

Derm irAEs offer an opportunity to elucidate some of the major unresolved questions concerning ICI cancer therapy. In general, derm irAEs do not lead to a discontinuation of ICI therapy: in our experience, only 6% of patients presented more severe grade 3 or 4 derm AEs. Derm irAEs are normally reversible and respond well to low doses of systemic steroids, provided there is accurate and early detection, and in exception for life-threatening toxic epidermal necrolysis or DRESS syndrome [44,45]. This finding could be related to a longer duration of treatment despite the appearance of derm irAEs.

Finally, we could not identify any independent marker associated with the emergence of irAEs, in the primary tumor or in the individual demographics, apart from the duration of treatment and baseline status, as discussed above. Another unresolved question involves the development of the late onset irAEs. Notably, in our series, most late onset irAEs were dermatological (11 out of 13 late onset events). In clinical practices, patients receive second line treatments after disease progression and, therefore, the principal causative drug is sometimes difficult to identify. Determining which patients are at higher risk of developing irAEs, even after therapy has been withdrawn, remains elusive.

The main limitations of this study include the retrospective methodology based on a single-center clinical setting allowing the inclusion of a limited sample of patients. The adverse events were all recategorized according to the CTCAE v.5.1, to have a homogeneous collection. The roles played by the specific drugs and therapeutic regimens used were not analyzed; moreover, the suspension of medication was not analyzed in terms of partial or definitive interruption, and the late onset irAEs could appear after the 3-month post-therapy follow-up. However, one strength of the present cohort is that only naïve patients were analyzed, avoiding the role of other therapies in the appearance of adverse events. In the same line, we did not assess the merging role of combined therapies during progression of the disease, such as immunotherapy plus radiotherapy or immunotherapy plus target therapy in the case of *BRAF* mutated tumors.

## 5. Conclusions

Dermatological immune-related adverse events (derm irAEs) during immunotherapy for advanced melanomas are the most frequent irAEs and tend to appear within the first 3 months of therapy. In view of our results, it is important to keep in mind that (1) any kind of toxicity could occur in the same patient at any time, during or after therapy. (2) The most severe derm irAEs and hepatobiliary irAEs could occur at any time, but tend to appear later than milder events, while gastrointestinal irAEs are more frequent during the first 2 months. (3) Patients who present derm irAEs can expect significantly better outcomes; however, this tendency is not statistically significant after adjusting for the clinical baseline status and the duration of the therapy.

Prospective large-scale studies, along with careful dermatological evaluation of derm irAEs, will help to elucidate which irAEs play relevant roles in predicting a better response.

## Figures and Tables

**Figure 1 cancers-14-01237-f001:**
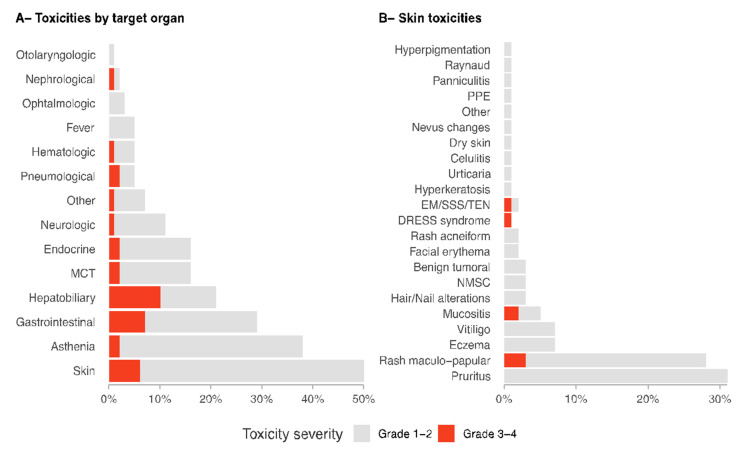
Details of toxicities box plots showing the percentage of patients who presented specific toxicities. Panel (**A**) shows the frequency of immune-related adverse events (irAEs) stratified by target organs. Fever and asthenia were considered as target symptoms due to the frequency of these side effects. Panel (**B**) shows dermatological irAEs in detail. Abbreviations: EM, erythema multiforme; MCT, musculoskeletal and connective tissue; PPE, palmar-plantar erythrodysesthesia; NMSC, non-melanoma skin cancer; SSS, Stevens-Johnson syndrome; TEN, toxic epidermal necrolysis.

**Figure 2 cancers-14-01237-f002:**
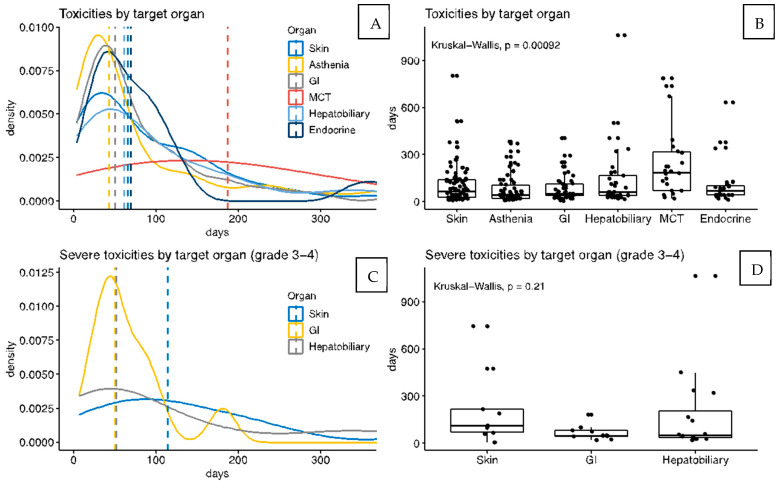
Density plot and boxplot toxicities. Box plots showing the percentages of patients who presented specific toxicities. Panel (**A**,**B**) show the frequency of immune-related adverse events (irAEs) stratified by target organs. Fever and asthenia were considered as target symptoms due to the frequency of these side effects. Panel (**C**,**D**) show the frequency of severe (grade 3–4) irAEs stratified by target organs. Abbreviations: EM, erythema multiforme; MCT, musculoskeletal and connective tissue; PPE, palmar-plantar erythrodysesthesia; NMSC, non-melanoma skin cancer; SSS, Stevens-Johnson syndrome; TEN, toxic epidermal necrolysis.

**Figure 3 cancers-14-01237-f003:**
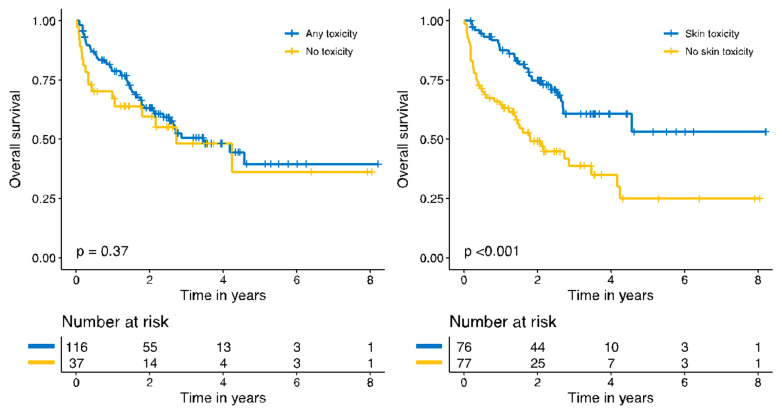
Kaplan–Meier plots for overall survival.

**Table 1 cancers-14-01237-t001:** Baseline characteristics of the cohort.

		All Toxicities			Skin Toxicities		
	Total	No toxicity	Toxicity	*p* Value	No Toxicity	Toxicity	*p* Value
	*N* = 153	*N = 37*	*N = 116*		*N = 77*	*N = 76*	
Age, Median *	60.0 [45.0;69.0]	60.0 [43.0;69.0]	61.0 [46.0;68.2]	0.309	62.0 [48.0;71.0]	56.5 [44.8;66.0]	0.171
Gender				0.348			0.575
Female	66 (43.1%)	13 (35.1%)	53 (45.7%)		31 (40.3%)	35 (46.1%)	
Male	87 (56.9%)	24 (64.9%)	63 (54.3%)		46 (59.7%)	41 (53.9%)	
AJCC stage				0.709			0.014
Stage IIIC	28 (18.9%)	5 (14.3%)	23 (20.4%)		9 (12.0%)	19 (26.0%)	
Stage IVA	22 (14.9%)	7 (20.0%)	15 (13.3%)		9 (12.0%)	13 (17.8%)	
Stage IVB	28 (18.9%)	7 (20.0%)	21 (18.6%)		12 (16.0%)	16 (21.9%)	
Stage IVC	70 (47.3%)	16 (45.7%)	54 (47.8%)		45 (60.0%)	25 (34.2%)	
LDH				0.427			0.079
Abnormal	22 (14.9%)	7 (20.6%)	15 (13.2%)		15 (20.8%)	7 (9.2%)	
Normal	126 (85.1%)	27 (79.4%)	99 (86.8%)		57 (79.2%)	69 (90.8%)	
Karnofsky score				0.063			0.055
>80	56 (56.0%)	9 (37.5%)	47 (61.8%)		25 (46.3%)	31 (67.4%)	
≤80	44 (44.0%)	15 (62.5%)	29 (38.2%)		29 (53.7%)	15 (32.6%)	
BRAF status:				0.844			0.053
BRAF mutated	37 (24.2%)	8 (21.6%)	29 (25.0%)		13 (16.9%)	24 (31.6%)	
Wild type	116 (75.8%)	29 (78.4%)	87 (75.0%)		64 (83.1%)	52 (68.4%)	
Melanoma location				0.598			0.277
Trunk	57 (37.5%)	13 (35.1%)	44 (38.3%)		23 (30.3%)	34 (44.7%)	
Lower limbs	25 (16.4%)	9 (24.3%)	16 (13.9%)		7 (9.2%)	7 (9.2%)	
Head and neck	19 (12.5%)	3 (8.1%)	16 (13.9%)		9 (11.8%)	10 (13.2%)	
Unknown	14 (9.2%)	3 (8.1%)	11 (9.6%)		17 (22.4%)	8 (10.5%)	
Acral	13 (8.6%)	2 (5.4%)	11 (9.6%)		6 (7.9%)	7 (9.2%)	
Upper limbs	13 (8.6%)	5 (13.5%)	8 (7.0%)		5 (6.6%)	6 (7.9%)	
Mucosa	11 (7.2%)	2 (5.4%)	9 (7.8%)		9 (11.8%)	4 (5.3%)	
Breslow index *	3.5 [2.0;6.0]	3.3 [1.9;6.0]	3.5 [2.2;6.0]	0.935	3.7 [2.0;6.6]	3.4 [2.0;4.8]	0.608
Ulceration				0.863			0.224
Absent	48 (38.4%)	11 (35.5%)	37 (39.4%)		20 (32.3%)	28 (44.4%)	
Present	77 (61.6%)	20 (64.5%)	57 (60.6%)		42 (67.7%)	35 (55.6%)	
Mitotic index *	5.0 [3.0;10.0]	6.0 [2.50;9.5]	5.0 [3.00;9.8]	0.974	5.0 [3.0;10.0]	5.5 [4.0;9.2]	0.726
Treatment duration *	126.0 [59;351]	69.0 [40;233]	164 [63;360]	0.012	66.0 [42;181]	312.0 [83;479]	<0.001

* Continuous variables expressed as median (IQR). Abbreviations: AJCC, American Joint Committee on Cancer; IQR, interquartile range; LDH lactate dehydrogenase.

**Table 2 cancers-14-01237-t002:** Univariate and multivariate analyses of overall survival.

Variables		HR (Univariable)	HR (Multivariable)
irAE	No skin toxicities	-	-
	Skin toxicities	0.40 (0.24–0.66, *p* < 0.001)	0.74 (0.44–1.25, *p* = 0.263)
AJCC Stage	IIIC	-	-
	IVA	1.12 (0.30–4.16, *p* = 0.870)	0.79 (0.21–2.97, *p* = 0.726)
	IVB	1.56 (0.49–4.92, *p* = 0.449)	1.02 (0.31–3.27, *p* = 0.980)
	IVC	6.46 (2.57–16.23, *p* < 0.001)	3.44 (1.32–8.99, *p* = 0.012)
Duration of treatment	<65 days	-	-
	65–331 days	0.53 (0.32–0.88, *p* = 0.014)	0.62 (0.37–1.06, *p* = 0.079)
	>331 days	0.11 (0.05–0.24, *p* < 0.001)	0.20 (0.08–0.48, *p* < 0.001)

Abbreviations: AJCC, American Joint Committee on Cancer; irAE, immune related adverse effect.

## Data Availability

The data presented in this study are available in this article.

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
