# Peer review of "Timeline of Adverse Events during Immune Checkpoint Inhibitors for Advanced Melanoma and Their Impacts on Survival"

_cancers, 2022, doi:10.3390/cancers14051237_

Round 1

Reviewer 1 Report

The authors have presented a paper about "Timeline of adverse events during immune checkpoint inhibitors for advanced melanoma and their impact on survival"

The topic itself is interesting however I ahve concerns I would like the authors to address as follows:

1) Figure 1 (A and B) are faded, please provide a higher quality image

2) The authors state that "adverse reaction were classified according to the organ/system and the degree of affectation according to the Common Terminology Criteria for Adverse Events of the National Cancer Institute Version 5.01(2018)" but the accrual period ranges between 2011 and 2019: were the adverse events recorded according to a previuos version of the CTCAE and then "covnerted"? Were they initailly recorded using another type of grading system? Since the focus of the paper is on side effects it of paramount importance to provide as many possbile details about this point. Please explain.

3) During a long followup patients may have undergone additional locoregional treatments (eg. radiotherapy): non mention about it is provided. If such data was not available the authors should state so, otherwise they should provide the % of patients who undergone additional treatments during immunotherapy

4) In the discussion section no mention is given with regard to the merging role of combination therapy (immunotherapy+radiotherapy); in particular no mention about the role of PIR (Peri-Induction Radiotherapy) amd PER (Post-Escape Radiotherapy) (see PMID: 33847208 for a detailed explantion). I believe it would be an addition to mention such therapeutic chance.

5) The manuscript would benefit from the addition of a clear explanation of the monitoring program followed by the authors (programmed visits, clinical evalutaion, imaging, lab exams and so on).

6) The authors state that "Interestingly, 13 adverse events (11.2%) occurred after discontinuing immunotherapy": it would be a good addition to provide a furhter discussion about the possible causes and also to have more information about the type of adverse events occurred.

Author Response

We really thank the reviewer for the time to review and improve our work. Following your suggestions, we have considered all your concerns as you can find below in the point to point answer: 

1) Figure 1 (A and B) are faded, please provide a higher quality image.

A) Totally agreed. We apologise for this, but the figures were of low quality just to upload in the submission process, the good quality ones are being sent now.  

2) The authors state that "adverse reaction were classified according to the organ/system and the degree of affectation according to the Common Terminology Criteria for Adverse Events of the National Cancer Institute Version 5.01(2018)" but the accrual period ranges between 2011 and 2019: were the adverse events recorded according to a previous version of the CTCAE and then "converted"? Were they initially recorded using another type of grading system? Since the focus of the paper is on side effects it of paramount importance to provide as many possible details about this point. Please explain.

A) Thanks for this consideration. Since the data was prospectively collected but the analysis was retrospective and as you notice, CTCAv5 was used. As a consequence, and just to consider the AEs in a homogeneous manner, some minor changes could have been necessary to update them to the latest CTCAv5. However, in our opinion this fact did not affect the evaluation of timeline or their role in favourable survival. We have included this clarification in the methodology and limitations.

3) During a long followup patients may have undergone additional locoregional treatments (eg. radiotherapy): non mention about it is provided. If such data was not available the authors should state so, otherwise they should provide the % of patients who undergone additional treatments during immunotherapy

A) We totally agree with you, and this is why we have excluded all patients who had received any other therapy, systemic or locoregional, including radiotherapy. As our main objective was to analysed the onset and impact of AEs of immunotherapy, only naive patients with no other therapy were analysed. We have highlighted this at the limitations section.

4) In the discussion section no mention is given with regard to the merging role of combination therapy (immunotherapy+radiotherapy); in particular no mention about the role of PIR (Peri-Induction Radiotherapy) amd PER (Post-Escape Radiotherapy) (see PMID: 33847208 for a detailed explantion). I believe it would be an addition to mention such therapeutic chance.

A) Thanks for your suggestion, very interesting point to be discussed. However, due to length limitations, we have just mentioned it at the end of the limitations and discussion.

5) The manuscript would benefit from the addition of a clear explanation of the monitoring program followed by the authors (programmed visits, clinical evalutaion, imaging, lab exams and so on).

A) Thanks for your suggestion, we have included a brief clarification about our surveillance protocol that follows the melanoma guidelines.

6) The authors state that "Interestingly, 13 adverse events (11.2%) occurred after discontinuing immunotherapy": it would be a good addition to provide a further discussion about the possible causes and also to have more information about the type of adverse events occurred.

A) The late onset of irAEs is a well stablished fact in the literature, likely due to a reinvigoration of immune system, the immune-mediated disorders can appear even after the CPI stopped. However, because of the retrospective analysis of our data, as we cannot assure right now if the late onset of adverse events were definitely caused by the immunotherapy or other ulterior drugs, we have removed this sentence to avoid misunderstanding.

Reviewer 2 Report

The authors performed an analysis of outcomes in patients receiving first line immunotherapy with specific focus on development of irAEs with most focus on dermatologic irAEs. They conclude that while dermatologic irAEs are common and associated with favorable clinical outcomes, this result doesn't bare true when correcting for duration of therapy and extent of disease. 

overall this is an interesting analysis. The tables/representation of the onset and frequency of various irAEs was interesting. 

The main areas of concern were in relation to grammar/English quality. Specific examples are below: 

First sentence of abstract—change “better” to improved or favorable

2nd line of abstract—“A total of 116 of 153 patients presented WITH at least one irAE.

Last sentence of abstract needs to be re-written for clarity

Line 50, colitis is mis-spelled

Author Response

Thanks for your revision.

We appreciate your corrections and as suggested, an English native editor has also improved our manuscript as you can see in the reviewed version. 

Round 2

Reviewer 1 Report

I have no further comments